# Lactate Neuroprotection against Transient Ischemic Brain Injury in Mice Appears Independent of HCAR1 Activation

**DOI:** 10.3390/metabo12050465

**Published:** 2022-05-21

**Authors:** Lara Buscemi, Melanie Price, Julia Castillo-González, Jean-Yves Chatton, Lorenz Hirt

**Affiliations:** 1Stroke Laboratory, Neurology Service, Department of Clinical Neurosciences, Lausanne University Hospital Centre and University of Lausanne, CH-1011 Lausanne, Switzerland; lara.buscemi@unil.ch (L.B.); melanie.price-hirt@chuv.ch (M.P.); juliacastillo@ipb.csic.es (J.C.-G.); 2Department of Fundamental Neurosciences, University of Lausanne, CH-1005 Lausanne, Switzerland; jean-yves.chatton@unil.ch; 3Departamento de Biología Celular e Inmunología, Instituto de Parasitología y Biomedicina López-Neyra (IPBLN-CSIC), 18016 Granada, Spain

**Keywords:** lactate, neuroprotection, MCAO, ischemia, stroke, HCAR1

## Abstract

Lactate can protect against damage caused by acute brain injuries both in rodents and in human patients. Besides its role as a metabolic support and alleged preferred neuronal fuel in stressful situations, an additional signaling mechanism mediated by the hydroxycarboxylic acid receptor 1 (HCAR1) was proposed to account for lactate’s beneficial effects. However, the administration of HCAR1 agonists to mice subjected to middle cerebral artery occlusion (MCAO) at reperfusion did not appear to exert any relevant protective effect. To further evaluate the involvement of HCAR1 in the protection against ischemic damage, we looked at the effect of HCAR1 absence. We subjected wild-type and HCAR1 KO mice to transient MCAO followed by treatment with either vehicle or lactate. In the absence of HCAR1, the ischemic damage inflicted by MCAO was less pronounced, with smaller lesions and a better behavioral outcome than in wild-type mice. The lower susceptibility of HCAR1 KO mice to ischemic injury suggests that lactate-mediated protection is not achieved or enhanced by HCAR1 activation, but rather attributable to its metabolic effects or related to other signaling pathways. Additionally, in light of these results, we would disregard HCAR1 activation as an interesting therapeutic strategy for stroke patients.

## 1. Introduction

Lactate, a metabolite of glycolysis, has beneficial effects in the recovery from damage caused by acute brain injury, as demonstrated in both preclinical animal models [1,2,3,4,5,6,7] and human patients [8,9,10]. However, the mechanisms by which the administration of exogenous lactate provides protection are yet to be unraveled. Since the discovery that lactate can act as an agonist of the hydroxycarboxylic acid receptor 1 (HCAR1) [11,12], expressed in rodent and human brain tissue [13,14,15,16,17], a dual mechanism of action for neuroprotection has been proposed [4]. On the one hand, lactate could be used as a preferential metabolic substrate providing energy to suffering neurons [18,19,20]. On the other, lactate could trigger a signaling response after binding to its receptor, a Gi-coupled protein receptor that modulates neuronal firing rates [13,15,21]. Indeed, an HCAR1 agonist has appeared to exert some protection in in vitro models of ischemia [4].

In previous work, we tested whether HCAR1 stimulation could provide neuroprotection in vivo by administering two different HCAR1 agonists to mice subjected to transient hypoxia–ischemia [22]. We evaluated the effect of the agonists on the ischemic lesion size and neurological outcome when given after reperfusion. However, the in vivo administration of the receptor agonists did not appear to exert any relevant protective effect, as previously seen with lactate administration. Although these results strongly suggested that the beneficial effects of exogenous lactate administration after MCAO could be mainly from a metabolic effect, we could not absolutely discard any neuroprotective effect of HCAR1 stimulation. Hence, to further ascertain whether the stimulation of HCAR1 could be considered as an interesting therapeutic option aimed at neuroprotection in acute brain injuries, we tested the effects of its absence in mice subjected to transient MCAO.

## 2. Results

To better understand the role of the lactate receptor HCAR1 in lactate-induced neuroprotection, we tested the outcomes of wild-type (WT) and HCAR1 knock-out (KO) mice subjected to transient MCAO and injected intravenously with either vehicle (PBS) or L-lactate (1 μmol/g) shortly after reperfusion, and considered the results in terms of lesion size (Figure 1) and functional impairment (Figure 2).

We observed a significant decrease in lesion size in HCAR1 KO animals compared to vehicle-treated WT animals (Figure 1A,B). As posterior brain structures appeared to experience damage to a lesser extent when WT mice were treated with lactate, as well as in both HCAR1 KO mice groups (Figure 1A), we analyzed the lesion areas on individual coronal sections (Figure 1C). The analysis confirmed a significant lesion-size reduction in the posterior slices of the HCAR1 KO animals compared to the vehicle-treated WT mice. The lesion size of the lactate-treated WT animals fell in between the larger lesions of WT PBS and KO mice, and was significantly smaller than that of the WT PBS mice on posterior slices (Figure 1C). A more detailed analysis of the damage on different brain structures showed smaller hippocampal, thalamic and midbrain lesions in vehicle-treated HCAR1 KO mice than in vehicle-treated WT mice, whereas no significant changes were observed in the striatum or the cortex (Figure 1D–H). Lactate-treated HCAR1 KO mice had significantly smaller hippocampal and thalamic damage compared to vehicle-treated WT mice. The analysis of the damage on white matter fiber tracts, which mainly affected tracts belonging to the lateral and medial forebrain bundle systems, showed a significantly larger damage in vehicle-treated WT mice compared to HCAR1 KO mice (Figure 1I). Although in lactate-treated WT animals the lesion size showed consistent decreasing trends in all of the structures analyzed, the differences did not reach statistical significance (*p* > 0.05).

After MCAO, a smaller proportion of lactate-treated WT and HCAR1 KO mice showed persisting circling behavior compared to vehicle-treated mice (Figure 2A). Vehicle-treated KO mice scored significantly better than vehicle-treated WT mice at 48 h (Figure 2A). 

We did not observe differences between the baseline performances of WT and HCAR1 KO mice in the rotarod task, and all mice reached similar maximal values of time on the rod after the training (Figure 2B). When subjected to the wire-hanging test training session, WT and KO mice showed similar performances (Figure 2C). 

After MCAO, both WT and KO vehicle-treated groups showed a slightly worsened performance on the rotarod task, whereas the two lactate-treated groups did not appear to be strongly affected by MCAO, and showed a better performance than vehicle-treated WT mice at 48 h (Figure 2D). 

We did not observe significant differences related either to the treatment or to the genotype for the wire-hanging test performance post-MCAO. However, in general, mice showed improvement with time, except for the lactate-treated KO mice (Figure 2E). 

## 3. Discussion

In the absence of HCAR1, the ischemic damage inflicted by transient MCAO was less pronounced, with smaller, more confined lesions on both gray and white matter and a better functional outcome than in WT mice. Of note, for both behavioral tests used in this study, HCAR1 KO mice did not show any relevant difference at baseline compared with age-matched WT mice, similarly to what was observed in a previous report with a plethora of behavioral tests [15]. Therefore, we could rule out a positive or negative contribution of the basal behavioral state of the mice in the post-ischemic outcome.

These results are in (partial) agreement with our previous report on the effect of HCAR1 agonists in the context of in vivo hypoxia–ischemia, where activation of the receptor failed to exert neuroprotection [22]. The lower susceptibility of the HCAR1 KO mice to the ischemic injury strengthens the idea that the activation of the receptor would not have a protective effect. Therefore, we do not consider that pursuing this approach would lead to an interesting therapeutic option.

Nevertheless, at present, how lactate exerts neuroprotection remains an open question. The discovery of HCAR1 [11,12], present in the brain [4,13,14,16,17], prompted the idea of a possible dual-protective effect of lactate in the context of stroke; a metabolic effect and a signaling effect mediated (possibly) by HCAR1 [4]. Lactate’s metabolic role in post-ischemic neuroprotection stands on several pieces of evidence. First, the conversion of lactate into a variety of brain metabolites when provided after an ischemic insult [6,23]. Second, the preferential use of lactate as an energy source following the recovery from hypoxic conditions [24]. Third, the loss of lactate protection after blocking its metabolism with oxamate, an LDH inhibitor [6]. Fourth, the insufficient protection provided by pyruvate [4,6,25]. Fifth, D-lactate, which can be metabolized, is protective [4]. Finally, the stronger protective effect exerted by lactate if given when normoxia and oxidative metabolism are restored [6]. 

Regarding the HCAR1-mediated signaling effect, the lack of protection following the administration of HCAR1 agonists in vivo [22], together with the better outcome of HCAR1 KO mice subjected to MCAO presented here, strongly questions the positive role of this receptor related to both endogenous and exogenous lactate effects. Although, neither result can entirely preclude a signaling role in lactate-mediated neuroprotection. Besides HCAR1, other lactate-sensitive receptors coupled to different signaling pathways appear to be expressed in different cell populations of the brain as well as in the periphery. In the brain, Gs-coupled receptors responsive to lactate have been postulated in the locus coeruleus [26], in the hippocampus [27], and in astrocytes [28,29]. In the periphery, other lactate receptors, such as GPR132, have been described in macrophages and are involved in the anti-inflammatory effects of lactate [30]. The lack of HCAR1 could, therefore, alter the role played by these and maybe other lactate-sensitive pathways, which could turn out to be beneficial for recovery from stroke. For instance, although there are reports suggesting HCAR1 involvement in the activation of the PI3K and ERK1/2 pathways by lactate [31], other reports show that lactate can activate these pathways independently of HCAR1 [32]. Additionally, HCAR1 has been shown to interact with other GPCRs, such as the adenosine A_1_ receptor, the GABA_B_ receptor and the α_2A_-adrenoreceptor, modifying their response [15]; therefore, the absence of HCAR1 could also affect the crosstalk with other receptors, indirectly modifying the response to the injury and, perhaps, the excitatory/inhibitory balance. Finally, as these are in vivo experiments that affect the whole animal, we cannot discard systemic effects related to the HCAR1 KO, as the lactate activation of HCAR1 inhibits lipolysis in adipose tissue [12] and has been related to blood pressure control [33]. Neither can we discard the possibility that systemic effects could be behind the apparent discrepancy between the responses to HCAR1 agonists observed in in vitro, ex vivo and in vivo experimental systems [4,14].

In light of these results, we consider that even though HCAR1 activation might not be considered as a promising pharmacological strategy to treat acute brain injuries such as stroke, it would be interesting to further investigate lactate’s mechanisms of neuroprotection, in particular, those related to its metabolic roles and how HCAR1 contributes to the ischemic injury.

## 4. Material and Methods

### 4.1. Transient Middle Cerebral Artery Occlusion (MCAO) Model

We performed all experiments in accordance with Swiss laws for the protection of animals with the approval of the Vaud Cantonal Veterinary Office. Mice had free access to food and water with housing on a 12 h light–dark cycle in a temperature- and humidity-controlled animal facility. For post-surgical recovery, mice were kept overnight in their home cages inside a temperature-controlled incubator at 28 °C. We used male wild-type C57BL/6N mice (40 animals, aged 8–12 weeks; Charles River, France) and HCAR1 KO mice (35 animals, aged 8–12 weeks; originally obtained from the Max Planck Institute for Heart and Lung Research [34] (Bad Nauheim, Germany) and kindly provided to us by Prof. J-Y. Chatton). Mice were subjected to transient ischemia using the intraluminal filament technique. Surgeries were performed under isoflurane anesthesia (1.5–2% in 70% N_2_O/30% O_2_) using a facemask. We monitored rectal temperature throughout surgery, which was maintained at 36.5 ± 0.5 °C, using a heating pad (FHC Inc., Bowdoin, ME, USA). We measured regional cerebral blood flow (rCBF) by laser-Doppler flowmetry (Perimed AB, Järfälla, Sweden) through a flexible probe fixed to the skull at 1 mm posterior and 6 mm lateral from the bregma. Transient focal ischemia was induced by introducing a silicon-coated nylon suture (Doccol Corp, Sharon, MA, USA) through the left common carotid artery into the internal left carotid artery and advancing it into the arterial circle to occlude the origin of the middle cerebral artery (MCA). We kept the occluding filament in place for 20 min, after which it was withdrawn to allow reperfusion. Surgeries were considered successful if rCBF during occlusion was <20% of the initial value, and reperfusion reached >50% of the initial value within 10 minutes of filament removal. Mice with satisfactory ischemia-reperfusion received a single intravenous (iv) injection of vehicle (5 μL/g PBS, pH 7.4) or sodium L-lactate (Fluka, Buchs, Switzerland, 5 μL/g of 200 mmol/L in PBS, pH 7.4), 15–20 min after reperfusion. We used a small group of mice to estimate the viability of the surgery on KO animals. These mice did not undergo behavioral assessment, but were included in the measurements of the lesion size. We randomized all treatments prior to the surgeries and behavioral testing, which were administered after reperfusion by an experimenter blinded to the treatment. We discarded animals that did not fulfill the inclusion criteria or did not receive a good intravenous injection (13 WT mice and 7 KO mice).

At 48 h after surgery, mice were sacrificed by intraperitoneal injection of 150 mg/kg sodium pentobarbital and intracardially perfused with 4% paraformaldehyde in PBS at pH 7.4. Brains were dissected, post-fixed overnight, cryoprotected in 30% sucrose, and cut into 25 μm cryostat coronal sections.

### 4.2. Ischemic Lesion Volume Determination

Cresyl violet-stained coronal cryostat sections (25 μm thick, 700 μm apart) were imaged under a light stereomicroscope (Nikon SMZ25). An examiner blinded to the genotype or treatment group measured lesion areas using the Allen Mouse Brain Atlas as a reference (http://mouse.brain-map.org/static/atlas, accessed on 28 March 2022) and ImageJ. The analysis of the damage on structures filed under the index *fiber tracts* in the Atlas is summarized as damage to white matter. We calculated the infarct volumes by multiplying the sum of the infarcted areas on each section by the spacing distance.

### 4.3. Functional Outcome Assessment

The functional outcome of mice was assessed by neurological score evaluation, rotarod test and wire-hanging test. LB and JCG performed the assessment and analysis of the behavioral outcomes and were blinded to the treatment. 

The neurological deficit was evaluated on all mice immediately after surgery, at 24 h and at 48 h after MCAO. Severity was graded using the following scale: (0) no observable deficit; (1) failure to extend the right forepaw; (1.5) intermittent circling; (2) persistent circling; (3) loss of righting reflex [35]. 

We subjected mice to the rotarod test, placing them on an accelerating rotating cylinder (Ugo Basile, Gemonio, Italy) and leaving them to run until they fell off the cylinder, completed two consecutive full turns gripping the rod or reached 300 s. The acceleration ramp was set to increase from 4 rpm to 40 rpm in 216 s and left at a constant 40 rpm speed until reaching 300 s from the start. The mice underwent three daily training trials for three consecutive days before surgery and three daily test trials at 24 h and 48 h after MCAO. The longest latency to fall from the rod over the three trials was scored for each mouse and time point and the percentage of the maximal performance at training was used as the outcome.

We also subjected mice to the wire-hanging test; they were suspended on a single wire stretched between two posts above soft ground, and allowed to escape towards the posts [22]. Mice were trained one day before intervention and scored for escaping (reaching the posts) and/or falling events at 24 h and 48 h after MCAO. The test lasted for a maximum of 180 s on the wire or a maximum of 10 falls. The overall performance was evaluated and scored as (from better to worse): only escape; neither escape nor fall; escape or fall; only fall. The Observer XT 14.0 software (Noldus, Wageningen, The Netherlands) was used for data extraction.

### 4.4. Statistical Analysis

We compared variables following Gaussian distributions with one-way analysis of variance (ANOVA) with Holm–Sidak’s post hoc test (multiple-group comparison against control). We compared non-Gaussian distributions with the Kruskal–Wallis test and Dunn’s post hoc test (multiple-group comparison against control). For the analysis of lesion size across the antero–posterior axis and of rotarod performance across time, we used two-way ANOVA with Holm–Sidak’s post hoc test (multiple-group comparison against control). Statistical tests were run on Graph-Pad Prism 8.0 (San Diego, CA, USA). On the box-and-whisker plots, the lines show the median and the 25th and 75th percentile; the whiskers correspond to the maximum and minimum values and each dot corresponds to an individual observation (animal). On the XY graphs, the error bars represent the standard error of the mean. On the scatter dot plots, the lines represent the mean and standard deviation and each dot corresponds to an individual observation (animal). We considered significance as *p* < 0.05.

## Figures and Tables

**Figure 1 metabolites-12-00465-f001:**
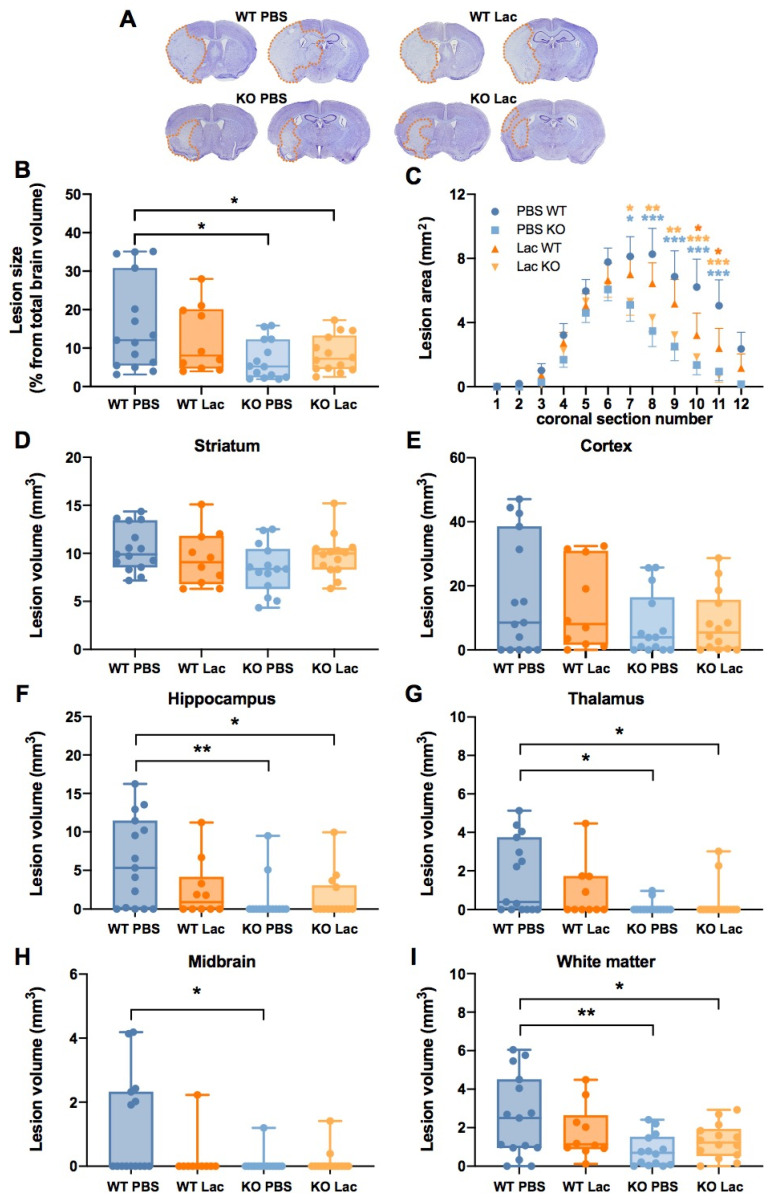
Effect of lactate on MCAO lesion size for WT and HCAR1 KO mice. (**A**) Representative images of cresyl violet-stained brain coronal sections of WT and HCAR1 KO mice subjected to transient MCAO and intravenously injected with PBS or lactate (Lac, 1 μmol/g) at reperfusion; lesions are outlined with a dashed orange line. Two images are provided for each condition, showing damage in the anterior (**left**) as well as posterior brain structures (**right**). (**B**) Total lesion size measurement. One-way ANOVA with Holm–Sidak’s post hoc test. (**C**) Measurement of the infarcted areas on individual coronal sections for the different conditions, showing that vehicle-treated WT mice suffered more extended damage. Two-way ANOVA with Holm–Sidak’s post hoc test. (**D**–**H**) Lesion volumes were measured in the following brain structures: (**D**) striatum; one-way ANOVA with Holm–Sidak’s post hoc test. (**E**) Cortex; Kruskal–Wallis with Dunn’s post hoc test. (**F**) Hippocampus; Kruskal–Wallis with Dunn’s post hoc test. (**G**) Thalamus; Kruskal–Wallis with Dunn’s post hoc test. (**H**) Midbrain; Kruskal–Wallis with Dunn’s post hoc test. (**I**) White matter; one-way ANOVA with Holm–Sidak’s post hoc test. * *p* < 0.05; ** *p* < 0.01; *** *p* < 0.001. In panel (**C**), the dark orange stars correspond to significant differences between WT PBS and WT Lac, light blue stars to significant differences between WT PBS and KO PBS and light orange stars to significant differences between WT PBS and KO lactate.

**Figure 2 metabolites-12-00465-f002:**
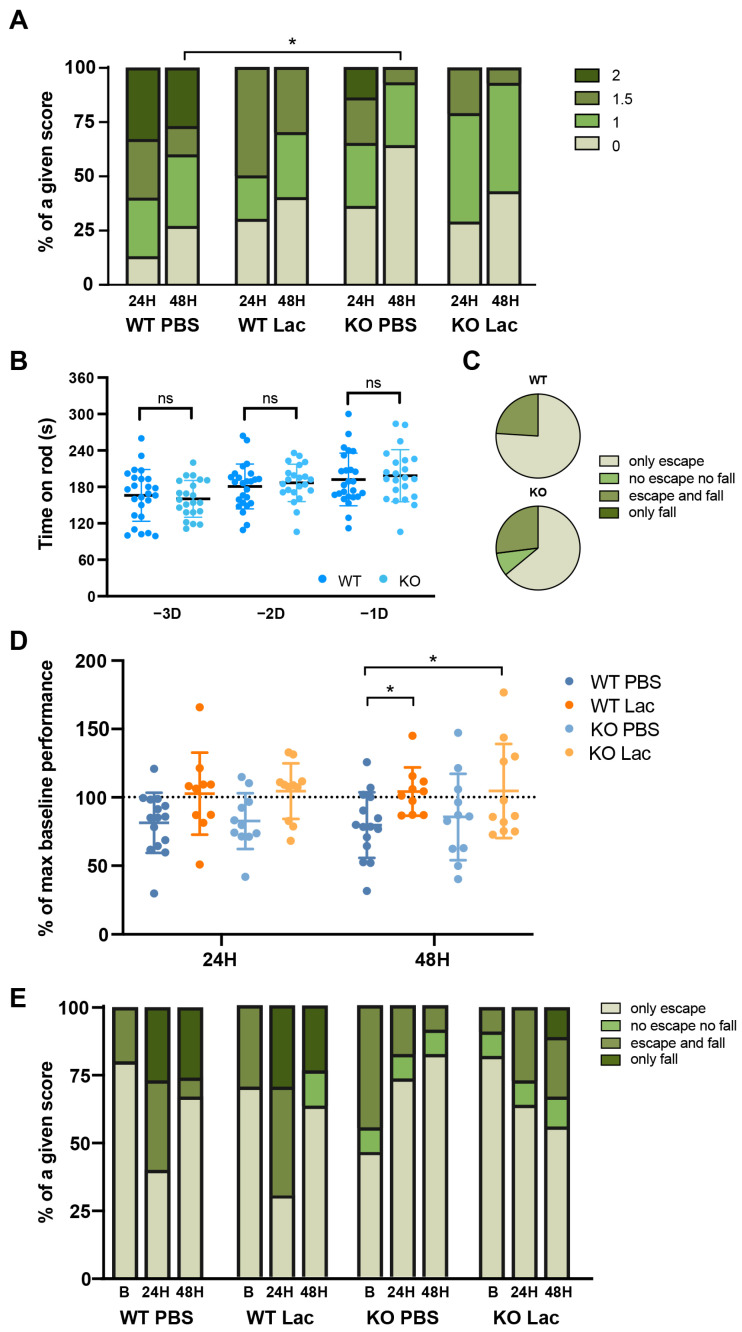
Effect of lactate on post-MCAO behavioral outcome of WT and HCAR1 KO mice. (**A**) Neurological deficit scores (0: no deficit; 1: failure to extend right forepaw; 1.5: intermittent circling; 2: persisting circling) evaluated at 24 h or 48 h after transient MCAO followed by treatment with PBS or lactate (Lac, 1 μmol/g) at reperfusion. Two-way ANOVA with Holm–Sidak’s post hoc test. (**B**) Time spent on the accelerating rod during the three consecutive training days (−3D, −2D, −1D) was similar for WT mice (dark blue) and KO mice (light blue). One-way ANOVA with Holm–Sidak’s post hoc test. (**C**) Wire-hanging test results (from better to worse: only escape; neither escape nor fall; escape and fall; only fall) obtained at baseline for each genotype. In the WT group, 76% of mice only escaped and 24% of mice escaped and fell, whereas in the KO group 64% of mice only escaped, 9% of mice neither escaped nor fell, and 27% of mice escaped and fell. (**D**) Percentage of the maximal performance achieved during the rotarod training of WT and HCAR1 KO mice tested at 24 h and 48 h after the surgery and subsequent treatment. Both WT and KO lactate-treated groups showed an improved performance at 48 h. Two-way ANOVA with Holm–Sidak’s post hoc test. (**E**) Scoring of the performance on the wire-hanging test at baseline (**B**), 24 h or 48 h after MCAO for each experimental condition. Two-way ANOVA with Holm–Sidak’s post hoc test. * *p* < 0.05.

## Data Availability

The data presented in this study are available on request from the corresponding author.

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
