# Peer review of "Lactate Neuroprotection against Transient Ischemic Brain Injury in Mice Appears Independent of HCAR1 Activation"

_metabolites, 2022, doi:10.3390/metabo12050465_

Round 1
Reviewer 1 Report
In this manuscript, authors investigated the neuroprotection of lactate on transient ischemic brain injury in MCAO mice model. They identified that the neuroprotection of lactate was independent of HCAR1 by ko mice. However, data from Figure 1 and 2 did not show the significant neuroprotection of lactate on the lesion size and behavioral outcome between WT PBS and WT lac. Indeed the neuroprotection effects were showed in HCAR1 ko mice. It will be better to check the downstream expression of signaling pathways to provide the mechanism.
Reviewer 2 Report
The manuscript titled as ‘Lactate neuroprotection against transient ischemic brain injury in mice appears independent of HCAR1 activation’ by Buscemi et al. tested the effects of the absence of the lactate receptor, HCAR1, in mice subjected to middle cerebral artery occlusion (MCAO). They found that in the absence of HCAR1, the ischemic damage inflicted by transient MCAO was less pronounced, with smaller, more confined lesions and a better functional outcome than in WT mice. This work should be of wide interests to most researchers on neuroscience and biomedicines.
This manuscript was well written, has some good findings, however, some results described need to be clarified further, for examples,
- The authors may need to assess whether HCAR1 knock-out mice exhibit persistent protection of grey and white matter after MCAO.
- The authors may need to assess any effects from lactate treatment grey and white matter after MCAO.
Reviewer 3 Report
This is an interesting and well-designed study exploring whether the lactate receptor HCAR1 could provide novel neuroprotective strategies in the ischemic brain, using a model of MCAO-reperfusion. The experiments are well planned and the data analysis is sound. The authors aimed to confirm their previous study, showing that HCAR1 agonists did not contribute to lactate neuroprotective effect after ischemia. Using a HCAR1-KO mouse, they show that although lactate is protective in the WT mouse, deletion of the HCAR1 receptor in itself is protective, independently of lactate, which suggests that HCAR1 could be detrimental to neuroprotection after MCAO. The analyses only compare the data from both HCAR1 KO groups and the WT lac group with the PBS WT group. It would be useful to address whether HCAR1 deletion provides a higher neuroprotection than lactate, by comparing WT lac with PBS KO, is that significant? Is the deletion of HCAR1 more protective that lactate? Would this mean that HCAR1 contributes somehow to ischemic damage independent of lactate or of receptor activation? As mentioned in the discussion, HCAR1 has been shown to associate with other GCPRs, such as the adenosine A1 receptor (A1R), the GABAB receptor (GABABR), and the α2A-adrenoreceptor (α2AR). Could HCAR1 modulate excitatory/inhibitory balance through these interactions? This effect of HCAR1 deletion on ischemic damage is the most important finding of this study and therefore the last sentence of the discussion should mention “it could be interesting to further investigate lactate’s mechanisms of neuroprotection, in particular those related to its metabolic roles and how HCAR1 contributes to ischemic injury”
Round 2
Reviewer 1 Report
The data of this manuscript had not improved after reversion. Based on the data of Figure 1, the whole infarct volume had no significant difference in WT PBS and WT LAC group although except some posterior sections. More new data need to added to prove the conclusion in this manuscript besides figure 1 and 2.
